# Benchmarking Tabular Foundation Models for Churn Prediction

**Sobhan Seyedzadeh** [1]   **Mostafa Karimi** [2]

## Abstract

Customer churn prediction is one of the most consequential ML-driven classification challenges across industries, where a single percentage-point gain in predictive accuracy translates into measurable retained revenue. Historically addressed by classical ML and, more recently, by deep learning and tree ensembles, churn prediction has lacked a systematic evaluation of the latest tabular foundation models (TFMs) built on in-context learning (ICL), a paradigm in which a model issues predictions from a labelled context in a single forward pass, with no retraining on the target dataset. We present the first cross-industry benchmark of sixteen models, spanning classical baselines, deep learning, tree ensembles, and six distinct TFMs (TabPFN, TabICL, TabDPT, Mitra, LimiX, and ConTextTab) across nine publicly available datasets covering seven industry sectors. TFMs collectively and decisively outperform all classical, deep learning, and tree-ensemble alternatives; TabICL v2, which operates without hardware constraints or subsampling, is the strongest overall performer, ranking in the top three on eight of nine datasets and outperforming XGBoost in PR-AUC on the majority of datasets (mean gain of about 3.3 pp; up to 9.23 pp on E-Commerce and 9.17 pp on IBM HR) with no dataset-specific retraining. These results position TFMs as strong, competitive candidates for churn prediction across sectors. SHAP and ablation analyses further identify the leading predictive driver of churn in each dataset studied, yielding an explainable end-to-end pipeline.

[*]Equal contribution [1]Independent Researcher [2]Amazon. Correspondence to: Sobhan Seyedzadeh <seyedzadeh.sobhan@gmail.com>.

*Proceedings of the $2^{nd}$ ICML Workshop on Foundation Models for Structured Data*, Seoul, South Korea. 2026. Copyright 2026 by the author(s).

## 1. Introduction

Growth strategies differ in risk. The Ansoff Matrix (Ansoff, 1957) identifies market penetration as the lowest-risk growth path, centred on retaining existing customers. Customer churn is the direct financial threat to this strategy: Accenture (2018) estimated $1 trillion of U.S. annual revenue at risk from customer switching, and Recurly (2024) estimated $129 billion lost annually to involuntary churn alone. At the firm level, even a single percentage-point improvement in churn identification can translate into meaningful retained revenue, which we quantify later under a conservative 30 percent retention-success assumption rather than assuming full recovery.

The literature on churn prediction has evolved along a clear progression. Classical methods including logistic regression, naïve Bayes, and SVMs, established transparent baselines but proved insufficient for non-linear, high-dimensional churn signals (De Caigny et al., 2018). Deep learning architectures offered theoretical expressiveness but have consistently underperformed on the small-to-medium tabular datasets that characterise churn problems, owing to data hunger and sensitivity to hyperparameter configuration. Tree ensemble methods, particularly XGBoost, addressed both limitations and emerged as the dominant practical baseline across industries (Ahmad et al., 2019; Lalwani et al., 2022). **Yet even XGBoost starts from scratch on every new dataset, with no transfer of knowledge across industries, and this is precisely the ceiling that tabular foundation models are designed to break.**

A new generation of TFMs, TabPFN v2 (Hollmann et al., 2025), TabICL v1 and v2 (Qu et al., 2025; 2026), TabDPT (Ma et al., 2025), Mitra (Zhang et al., 2025), LimiX (LimiX Team, 2025), and ConTextTab (Spinaci et al., 2025) has demonstrated that zero-shot transfer to new tabular datasets is achievable without retraining, through diverse architectural strategies spanning synthetic priors, real-data pretraining, and semantic column encoding. **To our knowledge, no prior study evaluates this full landscape of recent tabular foundation models for churn prediction, making this the first broad cross-industry TFM benchmark for the task.**

Three contributions follow: (1) the first cross-industry benchmark of sixteen models spanning classical baselines,

deep learning, tree ensembles, and six TFMs across nine publicly available datasets covering banking, fitness, e-commerce, telecommunications, healthcare, internet services, and HR; showing that TFMs outperform classical, deep-learning, and tree-ensemble families on the majority of datasets, with an accompanying estimate of financial impact; (2) TabICL v2 emerges as the strongest overall performer, ranking in the top three on eight of nine datasets while operating under no inference-time hardware constraints; no subsampling, no retrieval budget, no fine-tuning limit, making it the most practically deployable TFM evaluated; (3) SHAP- and EBM-validated ablation studies confirm the dominant churn driver in each industry sector, delivering an explainable, industry-agnostic end-to-end pipeline that converts churn scores into actionable retention priorities.

## 2. Background

### 2.1. Tabular Foundation Models

All TFMs in this benchmark share a common inference paradigm: given a labelled training set as context, each model issues predictions for held-out instances in a single forward pass, with no weight updates on the target dataset. This in-context learning (ICL) objective is realised by pre-training to minimise cross-entropy across a distribution of tabular datasets:

$$\mathcal{L}(\theta) = \mathbb{E}_{D \sim p(\cdot)}[-\log q_\theta(y_{\text{test}} \mid x_{\text{test}}, D_{\text{train}})] \quad (1)$$

Here, $q_\theta$ is the TFM parametrised by $\theta$, $D_{\text{train}}$ is the labelled context presented at inference, and $x_{\text{test}}$ is the held-out instance to classify. Six TFM families are evaluated; full architectural details and a comparative characteristics table are provided in Appendix C.

**TabPFN v2** (Hollmann et al., 2025) applies cell-level 2D attention over every (row, feature) pair, pre-trained on synthetic SCM/DAG priors. **TabICL** (Qu et al., 2025; 2026) introduces a two-stage column-then-row attention pipeline; v2 adds scalable QASSMax attention and repeated feature grouping, enabling processing of up to 500K rows without subsampling. **TabDPT** (Ma et al., 2025) pre-trains on 123 real-world datasets via self-supervised masking and selects the top-$K$ nearest training rows via FAISS at inference. **Mitra** (Zhang et al., 2025) uses a principled mixture of SCM and tree-based synthetic priors, optimised for diversity and distinctiveness across tasks. **LimiX** (LimiX Team, 2025) offers two scale variants (2M and 16M parameters) using asymmetric dual feature-level attention; 16M benefits from retrieval-augmented ensembling at larger contexts. **ConTextTab** (Spinaci et al., 2025) extends table-native ICL with lightweight semantic column-name embeddings via a frozen sentence encoder, pre-trained on the T4 real-world table corpus.

Furthermore, CARTE (Kim et al., 2024) and TabSTAR (Arazi et al., 2025) are architected for tables containing meaningful free-text columns (e.g., product names, clinical notes, CRM entries), making them well-suited to enterprise churn pipelines that incorporate unstructured interaction data; however, the nine datasets evaluated here contain no such columns, making a fair comparison impossible. Finally, TabArena (Erickson et al., 2025) is living benchmarking infrastructure; model selection in this study was guided by the TabArena leaderboard rankings.

### 2.2. Interpretability Methods

SHAP (Lundberg & Lee, 2017) assigns each feature an exact contribution score per prediction, guaranteed to sum to the model output, providing both global rankings and directional local explanations. LIME (Ribeiro et al., 2016) approximates local decision boundaries by perturbing inputs, making it model-agnostic but stochastic. EBM (Lou et al., 2012; Nori et al., 2019) is a transparent model by design, learning explicit per-feature contribution curves achievable without post-hoc tooling.

## 3. Experimental Setup

### 3.1. Datasets

Nine publicly available Kaggle datasets are used (Table 1). They span seven distinct industry sectors: banking, fitness, e-commerce, telecommunications, healthcare, internet services, and human resources, making this the broadest cross-industry evaluation of tabular foundation models for churn prediction to date. The datasets vary substantially in scale (1,470 to 72,274 samples), dimensionality (9 to 30 features), and churn rate (14.6% to 68.4%), collectively stress-testing each model across the full range of conditions encountered in real industry deployments.

*Table 1.* Dataset summary. $n$ = samples, $p$ = features, Churn% = minority class rate.

| Dataset | n | p | Churn% | Industry |
|---|---|---|---|---|
| Bank Customer Churn | 10,000 | 10 | 20.4% | Banking |
| Fitness Club Churn | 4,000 | 13 | 26.5% | Fitness |
| E-Commerce Churn | 5,630 | 18 | 16.8% | E-Commerce |
| IBM Telco Churn | 7,043 | 19 | 26.5% | Telecom |
| Telecommunications Churn | 2,666 | 19 | 14.6% | Telecom |
| Healthcare Patient Dropout | 2,000 | 18 | 68.4% | Healthcare |
| ISP Churn | 72,274 | 9 | 55.4% | Internet |
| Credit Card Customer Attr. | 10,127 | 19 | 16.1% | Banking |
| IBM HR Employee Attrition | 1,470 | 30 | 16.1% | HR |

Dataset links are provided in Appendix D.

## 3.2. Models and Protocol

Sixteen models: (1) *Foundation*: TabICL v2, TabICL v1, TabPFN v2, TabDPT, Mitra, LimiX-16M, LimiX-2M, ConTextTab; (2) *Neural*: Residual MLP, FT-Transformer; (3) *Tree Ensembles*: XGBoost, Random Forest; (4) *Classical*: Logistic Regression, SVM, KNN, Naive Bayes. EBM is evaluated separately in Section 5. All models use 5-fold stratified cross-validation; evaluation metrics are F1, ROC-AUC, PR-AUC, and Log Loss, where it penalises confident but incorrect predictions, making it a sensitive diagnostic for model calibration alongside discrimination metrics. In this investigation, PR-AUC is the primary metric throughout because churn datasets are class-imbalanced: missing a genuine churner carries a higher business cost than a false alarm. Baseline configuration: XGBoost, Random Forest, and the classical baselines were run with library-default hyperparameters; TFMs were used out-of-the-box with publicly released weights.

## 3.3. Hardware and Software

Experiments were run on Amazon SageMaker (AWS), instance `ml.g4dn.xlarge`, NVIDIA Tesla T4 GPU (16 GB VRAM), CUDA 12.4. Core packages: Python 3.11, PyTorch 2.6, scikit-learn 1.4, XGBoost 2.x, InterpretML (EBM), SHAP 0.44, LIME 0.2, pandas, and numpy. All TFMs use publicly released pre-trained weights; no custom training was performed; all usage was inference-only.

# 4. Results and Analysis

## 4.1. Main Results

Table 2 presents results for Datasets 1–3 (Bank, IBM Telco, ISP); the complete sixteen-model results across all nine datasets are in Tables A, B, C (Appendix A).

Figure 1 shows TabICL v2's PR-AUC advantage over XGBoost. **These results provide systematic evidence that tabular foundation models are competitive with, and frequently ahead of, strong tuned baselines for churn prediction.**

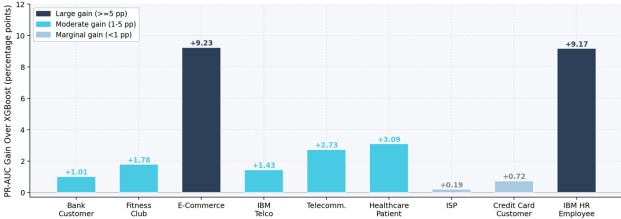

*Figure 1.* PR-AUC advantage of TabICL v2 over XGBoost (percentage points). E-Commerce (+9.23 pp) and IBM HR Employee Attrition (+9.17 pp) show the largest gains.

## 4.2. Discussions

Classical baselines provide a transparent starting point but expose a clear performance ceiling. Logistic Regression and Naive Bayes perform adequately where the churn signal is nearly linearly separable, yet fall sharply on datasets with non-linear interactions. FT-Transformer underperforms relative to both classical and tree-based alternatives across most industries, reinforcing a well-documented limitation of attention-based architectures on small tabular datasets. XGBoost emerges as the consistent non-foundation performance leader, outperforming all classical and deep learning alternatives in nearly every industry.

TabICL v2 outperforms XGBoost in PR-AUC on eight of nine datasets, with advantages reaching +9.23 pp (E-Commerce) and +9.17 pp (IBM HR). LimiX-16M achieves the highest ROC-AUC of all sixteen models on Bank Customer Churn (87.20%), while Mitra leads on Telecommunications Churn (F1 = 87.57%, PR-AUC = 89.52%).

Some inference-time constraints were encountered with some TFMs and are reported transparently. TabDPT used $K$=256 nearest-neighbour retrieval (vs. the default $K$=2,048) due to GPU memory limits, likely understating its full potential. LimiX variants ran without their retrieval ensemble and with a 40,000-row context cap on the ISP dataset. Mitra used a 2,048-sample context cap and a 300-second per-fold fine-tuning budget. ConTextTab exhibits strong discriminative ability (ROC-AUC 85–99 across datasets) but variable F1, with performance on minority-class detection correlated with the positive-class density within the 1,024-row context window. These are practical engineering trade-offs, not fundamental limitations of the models. **TabICL v2 operated under none of these restrictions**, processing every dataset in full in a single forward pass with fixed pre-trained weights, making it the most practically deployable TFM evaluated. Differences among the top TFMs are small and often within cross-validation variance, so TabICL v2's advantage is best read as operational, meaning it runs under no inference-time constraints, rather than as a statistically decisive metric lead.

Furthermore, all sixteen models cluster between 60–65% ROC-AUC on Healthcare (68.4% churn, $n$=2,000). This convergence indicates weak discriminative signal in the available features. The current features simply do not carry enough information to reliably distinguish churners from non-churners in this dataset. Adding clinically meaningful variables such as appointment frequency trends, billing history, or patient satisfaction scores which would likely unlock substantially stronger performance across all models.

**Datasets 1–3: Bank Customer Churn | IBM Telco Churn | ISP Churn**

| | Bank Customer Churn ($n$=10k, $p$=10, 20.4%) | | | | IBM Telco Churn ($n$=7,043, $p$=19, 26.5%) | | | | ISP Churn ($n$=72k, $p$=9, 55.4%) | | | |
|---|---|---|---|---|---|---|---|---|---|---|---|---|
| Model | F1 | ROC | PR | LL | F1 | ROC | PR | LL | F1 | ROC | PR | LL |
| *Tabular Foundation Models* | | | | | | | | | | | | |
| TabPFN v2 | 60.09(1.9) | 87.13(0.8) | 71.45(2.0) | 32.98(1.1) | 61.31(1.5) | **86.52(0.7)** | 69.21(1.0) | 39.28(1.0) | 94.65(0.1) | 98.27(0.5) | 98.75(0.0) | 16.26(0.2) |
| TabICL v2 | 59.76(1.9) | 87.12(0.7) | 71.50(2.0) | 32.94(1.1) | 61.46(1.1) | 86.51(0.8) | **69.32(1.4)** | 39.26(1.0) | **95.02(0.1)** | **98.56(0.0)** | **98.95(0.0)** | **14.97(0.2)** |
| TabICL v1 | 58.74(1.3) | 86.86(0.8) | 71.03(2.1) | 33.19(1.0) | 61.90(1.2) | 86.44(0.7) | 69.02(1.0) | 39.37(0.9) | 94.26(0.1) | 97.81(0.1) | 98.48(0.1) | 17.80(0.3) |
| TabDPT | 55.20(2.1) | 84.68(1.0) | 66.72(1.9) | 35.57(0.9) | 58.27(1.3) | 85.39(0.6) | 66.92(1.0) | 40.75(0.8) | 94.29(0.1) | 98.01(0.1) | 98.54(0.1) | 17.43(0.3) |
| Mitra | 57.96(1.7) | 85.98(0.8) | 69.17(2.1) | 34.02(1.0) | 60.62(1.8) | 86.05(0.6) | 67.26(1.9) | 39.84(0.8) | 94.08(0.1) | 97.18(0.4) | 98.09(0.2) | 19.72(1.0) |
| LimiX-16M | **60.28(1.9)** | **87.20(0.6)** | **71.64(1.9)** | **32.80(0.9)** | 60.04(1.6) | 86.41(0.7) | 69.05(1.1) | 39.33(0.9) | 93.98(0.1) | 98.27(0.0) | 98.74(0.0) | 17.06(0.2) |
| LimiX-2M | 60.10(1.8) | 87.19(0.8) | 71.48(2.1) | 32.85(1.1) | 61.41(0.9) | 86.51(0.7) | 69.24(0.9) | 39.27(1.0) | 84.51(1.1) | 97.73(0.1) | 98.33(0.1) | 26.18(0.9) |
| ConTextTab | 54.32(1.5) | 85.45(1.1) | 68.13(1.6) | 34.76(0.9) | 59.73(1.6) | 85.78(0.6) | 67.97(1.1) | 40.26(0.8) | 93.65(0.2) | 97.06(0.1) | 97.94(0.1) | 20.64(0.5) |
| *Neural Tabular Models* | | | | | | | | | | | | |
| Residual MLP | 58.95(2.5) | 84.87(1.1) | 67.20(2.4) | 45.91(3.5) | **62.71(1.7)** | 85.23(1.0) | 65.73(1.7) | 52.41(3.7) | 93.49(0.3) | 97.27(0.2) | 98.10(0.1) | 19.92(0.6) |
| FT-Transformer | 45.43(1.4) | 74.09(1.9) | 49.44(2.0) | 59.62(2.7) | 53.42(2.2) | 75.99(1.8) | 54.16(3.9) | 60.88(2.1) | 92.89(0.4) | 96.31(0.1) | 97.43(0.1) | 22.84(0.6) |
| *Tree Ensembles* | | | | | | | | | | | | |
| XGBoost | 59.21(2.1) | 86.59(0.7) | 70.49(1.8) | 33.40(0.8) | 59.59(2.4) | 85.74(0.9) | 67.89(2.1) | 40.30(1.2) | 94.72(0.1) | 98.26(0.1) | 98.76(0.0) | 16.15(0.3) |
| Random Forest | 58.37(2.2) | 85.56(0.5) | 68.26(2.0) | 35.15(1.0) | 57.11(1.8) | 83.82(0.8) | 63.41(2.2) | 45.82(3.4) | 94.51(0.1) | 98.16(0.0) | 98.62(0.1) | 17.78(0.7) |
| *Classical Baselines* | | | | | | | | | | | | |
| Log. Regression | 35.98(1.4) | 77.45(1.6) | 51.35(3.1) | 42.01(1.1) | 61.91(2.1) | 85.71(0.9) | 67.37(1.1) | 40.32(1.2) | 85.48(0.2) | 90.10(0.2) | 91.82(0.2) | 40.38(0.2) |
| SVM | 53.28(1.9) | 82.00(0.9) | 65.81(2.6) | 37.21(1.2) | 51.51(1.2) | 79.12(0.8) | 59.54(1.7) | 47.57(0.6) | 84.43(0.3) | 90.22(0.2) | 92.28(0.2) | 39.50(0.4) |
| KNN | 45.74(2.4) | 83.25(0.7) | 61.96(2.1) | 54.35(6.7) | 49.64(2.2) | 78.51(1.5) | 57.39(1.7) | 79.18(12.6) | 82.16(0.1) | 89.07(0.2) | 91.19(1.5) | 55.77(2.9) |
| Naive Bayes | 36.51(1.1) | 77.79(1.5) | 48.91(1.9) | 42.22(1.1) | 61.57(1.3) | 82.27(1.0) | 60.44(2.2) | 135.45(7.7) | 82.34(0.2) | 88.12(0.2) | 90.70(0.1) | 39.71(0.3) |

*Table 2.* Main results for Datasets 1–3: Bank Customer Churn, IBM Telco Churn, and ISP Churn. Values are mean% (std) under 5-fold stratified cross-validation. **Bold** = best; underline = second-best.

### 4.3. Financial Impact

Table 3 translates observed PR-AUC gains into business value. Assuming a conservative 30% retention success rate on additionally recovered churners, TabICL v2 retains approximately 26 additional customers per year on the E-Commerce dataset alone, with no change to product or pricing.

*Table 3.* Financial impact of TabICL v2 vs. XGBoost.

| Dataset | PR-AUC Gap | Annual Churners | Est. Retained (30%) | Revenue Impact |
|---|---|---|---|---|
| E-Commerce ($n$=5,630) | +9.23 pp | ∼946 | ∼26 | ∼$7,800 at $300/cust. |
| Fitness Club ($n$=4,000) | +1.78 pp | ∼1,060 | ∼6 | ∼$1,800 at $300/memb. |
| IBM Telco ($n$=7,043) | +1.43 pp | ∼1,868 | ∼8 | ∼$3,200 at $400/cust. |
| Bank Customer ($n$=10,000) | +1.01 pp | ∼2,040 | ∼6 | ∼$3,000 at $500/cust. |

## 5. Interpretability Analysis

Industry adoption demands more than predictive performance. Analysts using XGBoost today rely on SHAP to understand feature contributions; the practical question is whether a higher-capacity TFM can be made equally interpretable. A complementary tradition exists in EBM, which learns explicit per-feature contribution curves that any stakeholder can inspect without post-hoc tooling.

Our benchmark establishes TabICL v2 as the performance leader across industries. Applied post-hoc, SHAP recovers the same dominant risk features that EBM identifies natively, but paired with materially stronger predictive performance. SHAP and LIME agree on the top feature in 7 of 9 datasets; where they disagree (Fitness Club, Telecommunications), ablation confirms SHAP's selection caused

the larger performance drop. EBM correctly identified the ablation-confirmed top feature in 8 of 9 datasets.

### 5.1. Ablation Study

Removing the top two SHAP-identified features drops TabICL v2 by up to 17.0 pp ROC-AUC and 19.7 pp F1 (Bank Customer Churn), confirming these are the dominant predictive drivers. This is a standard SHAP validation: features ranked important should, when withheld, produce measurable degradation. Full ablation results with industry-by-industry business implications are in Table D (Appendix B). These findings translate directly into actionable retention priorities for industry practitioners: each dataset's dominant predictive driver of churn is empirically validated and ready to inform campaign targeting.

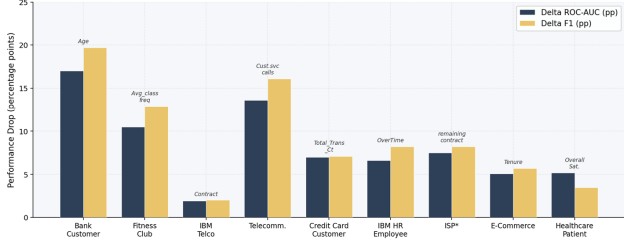

*Figure 2.* Ablation: TabICL v2 performance drop (pp) when both top SHAP features are removed, confirming their predictive importance.

## 6. Conclusion

Customer churn prediction is now a tractable problem for modern tabular models, and practitioners can approach it

with a stronger, better-understood toolkit. Across nine real-world datasets and seven industry sectors, the evidence establishes three durable conclusions. First, **tabular foundation models, led by TabICL v2, are strong and competitive tools for churn prediction**: they outperform all classical, deep learning, and tree-ensemble alternatives without industry-specific retraining, with TabICL v2 ranking in the top three on eight of nine datasets and operating under no inference-time hardware constraints. Second, the **business impact is quantifiable and material**. Third, **the actionable insights are validated**: SHAP explanations, confirmed by ablation, identify the dominant churn driver in each industry sector, giving analysts an empirically grounded basis for retention campaign design.

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

# A. Full Performance Results

Tables A, B, C reports all sixteen models across nine datasets under 5-fold stratified cross-validation.

*Table A.* Values are mean% with standard deviation in parentheses. **Bold** = best; underline = second-best per metric per dataset. ROC = ROC-AUC; PR = PR-AUC; LL = Log Loss.

## Datasets 1–3: Bank Customer Churn | IBM Telco Churn | ISP Churn

| Model | Bank Customer Churn (n=10k, p=10, 20.4%) | | | | IBM Telco Churn (n=7,043, p=19, 26.5%) | | | | ISP Churn (n=72k, p=9, 55.4%) | | | |
|---|---|---|---|---|---|---|---|---|---|---|---|---|
| | F1 | ROC | PR | LL | F1 | ROC | PR | LL | F1 | ROC | PR | LL |
| *Tabular Foundation Models* | | | | | | | | | | | | |
| TabPFN v2 | 60.09(1.9) | 87.13(0.8) | 71.45(2.0) | 32.98(1.1) | 61.31(1.5) | **86.52(0.7)** | 69.21(1.0) | 39.28(1.0) | 94.65(0.1) | 98.27(0.5) | 98.75(0.0) | 16.26(0.2) |
| TabICL v2 | 59.76(1.9) | 87.12(0.7) | 71.50(2.0) | 32.94(1.1) | 61.46(1.1) | 86.51(0.8) | **69.32(1.4)** | **39.26(1.0)** | **95.02(0.1)** | **98.56(0.0)** | **98.95(0.0)** | **14.97(0.2)** |
| TabICL v1 | 58.74(1.3) | 86.86(0.8) | 71.03(2.1) | 33.19(1.0) | 61.90(1.2) | 86.44(0.7) | 69.02(1.0) | 39.37(0.9) | 94.26(0.1) | 97.81(0.1) | 98.48(0.1) | 17.80(0.3) |
| TabDPT | 55.20(2.1) | 84.68(1.0) | 66.72(1.9) | 35.57(0.9) | 58.27(1.3) | 85.39(0.6) | 66.92(1.0) | 40.75(0.8) | 94.29(0.1) | 98.01(0.1) | 98.54(0.1) | 17.43(0.3) |
| Mitra | 57.96(1.7) | 85.98(0.8) | 69.17(2.1) | 34.02(1.0) | 60.62(1.8) | 86.05(0.6) | 67.26(1.9) | 39.84(0.8) | 94.08(0.1) | 97.18(0.4) | 98.09(0.2) | 19.72(1.0) |
| LimiX-16M | **60.28(1.9)** | **87.20(0.6)** | **71.64(1.9)** | **32.80(0.9)** | 60.04(1.6) | 86.41(0.7) | 69.05(1.1) | 39.33(0.9) | 93.98(0.1) | 98.27(0.0) | 98.74(0.0) | 17.06(0.2) |
| LimiX-2M | 60.10(1.8) | 87.19(0.8) | 71.48(2.1) | 32.85(1.1) | 61.41(0.9) | 86.51(0.7) | 69.24(0.9) | 39.27(1.0) | 84.51(1.1) | 97.73(0.1) | 98.33(0.1) | 26.18(0.9) |
| ConTextTab | 54.32(1.5) | 85.45(1.1) | 68.13(1.6) | 34.76(0.9) | 59.73(1.6) | 85.78(0.6) | 67.97(1.1) | 40.26(0.8) | 93.65(0.2) | 97.06(0.1) | 97.94(0.1) | 20.64(0.5) |
| *Neural Tabular Models* | | | | | | | | | | | | |
| Residual MLP | 58.95(2.5) | 84.87(1.1) | 67.20(2.4) | 45.91(3.5) | **62.71(1.7)** | 85.23(1.0) | 65.73(1.7) | 52.41(3.7) | 93.49(0.3) | 97.27(0.2) | 98.10(0.1) | 19.92(0.6) |
| FT-Transformer | 45.43(1.4) | 74.09(1.9) | 49.44(2.0) | 59.62(2.7) | 53.42(2.2) | 75.99(1.8) | 54.16(3.9) | 60.88(2.1) | 92.89(0.4) | 96.31(0.1) | 97.43(0.1) | 22.84(0.6) |
| *Tree Ensembles* | | | | | | | | | | | | |
| XGBoost | 59.21(2.1) | 86.59(0.7) | 70.49(1.8) | 33.40(0.8) | 59.59(2.4) | 85.74(0.9) | 67.89(2.1) | 40.30(1.2) | 94.72(0.1) | 98.26(0.1) | 98.76(0.0) | 16.15(0.3) |
| Random Forest | 58.37(2.2) | 85.56(0.5) | 68.26(2.0) | 35.15(1.0) | 57.11(1.8) | 83.82(0.8) | 63.41(2.2) | 45.82(3.4) | 94.51(0.1) | 98.16(0.0) | 98.62(0.1) | 17.78(0.7) |
| *Classical Baselines* | | | | | | | | | | | | |
| Log. Regression | 35.98(1.4) | 77.45(1.6) | 51.35(3.1) | 42.01(1.1) | 61.91(2.1) | 85.71(0.9) | 67.37(1.1) | 40.32(1.2) | 85.48(0.2) | 90.10(0.2) | 91.82(0.2) | 40.38(0.2) |
| SVM | 53.28(1.9) | 82.00(0.9) | 65.81(2.6) | 37.21(1.2) | 51.51(1.2) | 79.12(0.8) | 59.54(1.7) | 47.57(0.6) | 84.43(0.3) | 90.22(0.2) | 92.28(0.2) | 39.50(0.4) |
| KNN | 45.74(2.4) | 83.25(0.7) | 61.96(2.1) | 54.35(6.7) | 49.64(2.2) | 78.51(1.5) | 57.39(1.7) | 79.18(12.6) | 82.16(0.1) | 89.07(0.2) | 91.19(1.5) | 55.77(2.9) |
| Naive Bayes | 36.51(1.1) | 77.79(1.5) | 48.91(1.9) | 42.22(1.1) | 61.57(1.3) | 82.27(1.0) | 60.44(2.2) | 135.45(7.7) | 82.34(0.2) | 88.12(0.2) | 90.70(0.1) | 39.71(0.3) |

TabPFN v2 on ISP: 20-model ensemble with 10k subsamples each. ConTextTab on ISP: 1,024-row stratified context per fold.

*Table B.* Values are mean% with standard deviation in parentheses. **Bold** = best; underline = second-best per metric per dataset. ROC = ROC-AUC; PR = PR-AUC; LL = Log Loss.

## Datasets 4–6: Fitness Club Churn | Telecommunications Churn | Credit Card Attrition

| Model | Fitness Club Churn (n=4,000, p=13, 26.5%) | | | | Telecom Churn (n=2,666, p=19, 14.6%) | | | | Credit Card Attrition (n=10,127, p=19, 16.1%) | | | |
|---|---|---|---|---|---|---|---|---|---|---|---|---|
| | F1 | ROC | PR | LL | F1 | ROC | PR | LL | F1 | ROC | PR | LL |
| *Tabular Foundation Models* | | | | | | | | | | | | |
| TabPFN v2 | 91.91(1.3) | **99.15(0.2)** | **98.13(0.4)** | **9.83(1.3)** | 84.95(2.8) | 91.92(0.9) | 88.24(2.1) | 15.24(1.8) | **92.98(0.7)** | **99.54(0.1)** | 97.83(0.5) | **6.16(0.8)** |
| TabICL v2 | 92.39(1.4) | 99.14(0.3) | 98.12(0.5) | 9.85(1.5) | 86.72(1.5) | **93.10(0.9)** | 89.40(1.4) | 13.19(1.3) | 92.11(1.0) | 99.53(0.1) | 97.83(0.5) | 6.30(1.0) |
| TabICL v1 | 91.84(1.6) | 99.01(0.3) | 97.83(0.7) | 10.67(1.7) | 77.69(1.8) | 91.71(1.6) | 84.36(3.4) | 18.07(2.0) | 91.16(1.5) | 99.32(0.3) | 96.83(0.5) | 7.62(0.8) |
| TabDPT | 91.42(0.9) | 98.74(0.4) | 97.46(0.6) | 11.79(1.2) | 71.05(3.0) | 90.36(1.7) | 81.41(2.6) | 20.71(1.0) | 84.36(2.0) | 98.04(0.4) | 92.58(1.1) | 12.83(0.9) |
| Mitra | 92.34(0.9) | 99.05(0.3) | 97.95(0.6) | 10.30(1.3) | **87.57(1.0)** | 92.92(1.8) | **89.52(1.7)** | **12.39(1.2)** | 89.28(1.4) | 99.11(0.2) | 95.93(0.9) | 8.79(0.9) |
| LimiX-16M | 92.29(1.0) | 99.12(0.3) | 98.12(0.6) | 9.96(1.6) | 83.02(2.7) | 91.28(1.4) | 87.37(2.8) | 16.45(2.0) | 92.15(0.8) | 99.49(0.1) | 97.59(0.5) | 6.55(0.8) |
| LimiX-2M | **92.55(0.9)** | 99.11(0.4) | 98.10(0.5) | 9.96(1.5) | 85.33(3.6) | 92.07(1.7) | 88.28(2.2) | 14.98(2.0) | 92.84(0.9) | 99.54(0.1) | **97.85(0.4)** | 6.20(0.7) |
| ConTextTab | 91.93(1.6) | 99.00(0.4) | 97.87(0.8) | 10.57(2.1) | 72.31(4.1) | 90.54(1.1) | 80.59(3.0) | 20.62(1.2) | 90.55(1.4) | 99.21(0.2) | 96.50(0.6) | 8.22(0.8) |
| *Neural Tabular Models* | | | | | | | | | | | | |
| Residual MLP | 85.81(3.2) | 97.76(0.7) | 94.93(2.1) | 21.24(3.4) | 59.60(3.6) | 86.37(1.9) | 66.88(4.3) | 38.22(2.2) | 77.93(1.6) | 97.30(0.5) | 88.85(2.0) | 20.69(1.8) |
| FT-Transformer | 70.35(1.7) | 90.09(1.0) | 76.02(1.5) | 41.25(2.2) | 33.73(3.2) | 69.93(4.1) | 28.96(2.8) | 64.44(4.7) | 48.13(1.4) | 82.02(1.4) | 49.30(2.1) | 51.74(3.3) |
| *Tree Ensembles* | | | | | | | | | | | | |
| XGBoost | 89.00(2.1) | 98.35(0.4) | 96.34(0.9) | 14.07(1.7) | 82.98(3.3) | 91.46(0.7) | 86.67(2.4) | 17.01(1.5) | 90.98(1.5) | 99.37(0.2) | 97.11(0.9) | 7.35(1.0) |
| Random Forest | 84.33(2.1) | 97.23(0.5) | 92.99(1.6) | 19.87(1.4) | 78.14(1.7) | 90.63(1.7) | 86.22(1.8) | 21.34(0.8) | 85.50(1.6) | 98.76(0.3) | 94.64(1.1) | 13.27(0.5) |
| *Classical Baselines* | | | | | | | | | | | | |
| Log. Regression | 85.87(2.1) | 97.65(0.5) | 94.78(1.0) | 16.86(1.5) | 31.55(2.7) | 80.05(3.0) | 45.35(7.2) | 33.70(2.3) | 60.59(2.8) | 90.90(0.6) | 70.78(1.0) | 25.80(0.6) |
| SVM | 47.04(4.4) | 92.78(0.9) | 82.90(1.8) | 31.26(2.1) | 47.76(4.6) | 76.48(4.1) | 54.80(7.8) | 32.60(2.6) | 48.29(3.0) | 83.57(1.2) | 53.36(1.9) | 33.82(0.9) |
| KNN | 75.79(2.5) | 91.96(1.3) | 79.89(2.4) | 47.83(9.5) | 27.74(4.9) | 68.40(2.9) | 41.27(4.3) | 96.52(18.4) | 61.97(1.8) | 90.94(1.0) | 68.15(1.7) | 48.57(9.1) |
| Naive Bayes | 55.81(4.5) | 79.05(1.9) | 65.89(3.7) | 44.15(2.4) | 32.31(3.5) | 68.72(3.4) | 32.14(3.4) | 42.26(2.5) | 43.19(2.9) | 70.91(2.5) | 33.43(2.8) | 43.81(2.1) |

TabPFN v2 on ISP: 20-model ensemble with 10k subsamples each. ConTextTab on ISP: 1,024-row stratified context per fold.

*Table C.* Values are mean% with standard deviation in parentheses. **Bold** = best; underline = second-best per metric per dataset. ROC = ROC-AUC; PR = PR-AUC; LL = Log Loss.

### Datasets 7–9: E-Commerce Churn | Healthcare Patient Dropout | IBM HR Attrition

| | E-Commerce Churn (n=5,630, p=18, 16.8%) | | | | Healthcare Dropout (n=2,000, p=18, 68.4%) | | | | IBM HR Attrition (n=1,470, p=30, 16.1%) | | | |
|---|---|---|---|---|---|---|---|---|---|---|---|---|
| **Model** | F1 | ROC | PR | LL | F1 | ROC | PR | LL | F1 | ROC | PR | LL |
| *Tabular Foundation Models* | | | | | | | | | | | | |
| TabPFN v2 | 94.07(2.0) | 99.69(0.1) | 98.68(0.5) | 5.72(0.9) | 81.00(0.6) | 63.87(2.9) | 78.68(1.8) | 60.01(1.0) | 48.89(3.0) | 84.10(2.9) | 64.78(5.7) | 31.44(2.3) |
| TabICL v2 | **97.27(1.5)** | **99.93(0.1)** | **99.70(0.3)** | **2.40(1.1)** | 80.88(0.8) | **64.96(2.7)** | **79.64(1.6)** | **59.63(1.2)** | **57.21(4.8)** | **85.57(3.0)** | **68.15(5.2)** | **29.64(2.8)** |
| TabICL v1 | 93.69(2.4) | 99.66(0.1) | 98.52(0.6) | 5.56(1.3) | 81.06(0.5) | 64.30(3.4) | 78.89(1.6) | 59.77(1.2) | 51.02(6.1) | 84.44(2.8) | 64.08(6.2) | 31.44(2.3) |
| TabDPT | 97.26(1.3) | 99.91(0.1) | 99.61(0.3) | 3.10(1.0) | 80.48(0.5) | 60.71(2.0) | 76.62(2.2) | 61.38(1.0) | 37.35(6.1) | 80.50(3.3) | 56.95(8.2) | 34.37(2.3) |
| Mitra | 76.07(3.4) | 94.46(1.2) | 83.93(3.0) | 20.18(1.9) | 80.49(0.4) | 64.37(3.0) | 78.81(2.3) | 59.99(1.1) | 50.53(2.5) | 84.79(3.4) | 63.77(6.6) | 31.18(3.2) |
| LimiX-16M | 91.08(1.7) | 99.41(0.2) | 97.51(0.8) | 9.59(1.1) | 80.98(0.5) | 64.36(2.8) | 78.76(1.7) | 59.76(1.0) | 49.25(4.2) | 84.44(2.6) | 65.58(6.6) | 31.25(2.4) |
| LimiX-2M | 93.42(1.8) | 99.47(0.2) | 97.72(0.7) | 8.25(0.9) | 80.73(0.5) | 63.97(2.7) | 78.87(1.8) | 59.90(1.1) | 52.02(3.1) | 84.99(2.5) | 66.73(5.2) | 30.77(2.1) |
| ConTextTab | 72.84(2.9) | 94.40(0.9) | 83.62(2.0) | 20.91(1.3) | **81.37(0.3)** | 64.41(2.7) | 79.06(2.1) | 59.75(0.7) | 43.26(5.8) | 83.88(2.8) | 63.34(7.2) | 31.85(2.1) |
| *Neural Tabular Models* | | | | | | | | | | | | |
| Residual MLP | 83.20(7.6) | 97.84(1.3) | 91.64(5.7) | 18.79(8.4) | 60.95(8.9) | 60.92(1.4) | 76.43(1.5) | 71.99(5.4) | 48.37(2.5) | 78.54(2.0) | 54.76(4.9) | 56.45(12.1) |
| FT-Transformer | 54.69(5.1) | 83.86(2.8) | 57.03(7.1) | 48.85(4.8) | 52.84(33.3) | 53.00(3.1) | 71.78(2.9) | 68.59(3.0) | 39.22(2.1) | 72.30(2.4) | 41.70(6.5) | 60.89(5.1) |
| *Tree Ensembles* | | | | | | | | | | | | |
| XGBoost | 81.79(3.3) | 97.43(0.5) | 90.47(1.7) | 15.51(1.2) | 78.11(2.1) | 61.24(4.9) | 76.55(3.0) | 64.77(4.3) | 45.04(4.5) | 82.23(1.8) | 58.98(3.9) | 35.54(2.2) |
| Random Forest | 88.55(2.7) | 99.05(0.4) | 96.54(1.0) | 14.06(1.1) | 81.08(0.6) | 63.14(2.4) | 77.33(1.5) | 60.21(0.7) | 29.80(5.4) | 81.36(3.0) | 57.53(8.4) | 34.86(1.7) |
| *Classical Baselines* | | | | | | | | | | | | |
| Log. Regression | 62.53(1.0) | 89.06(0.4) | 71.08(1.7) | 28.16(0.6) | 79.42(0.6) | 64.31(2.5) | 79.13(1.9) | 60.04(1.1) | 35.41(5.9) | 78.06(4.3) | 49.96(9.5) | 36.86(3.7) |
| SVM | 61.57(1.1) | 88.67(0.7) | 69.35(2.1) | 28.92(0.9) | 81.21(0.2) | 60.70(1.8) | 76.14(1.6) | 61.01(0.4) | 0.00(0.0) | 48.13(3.8) | 17.07(1.5) | 44.58(0.5) |
| KNN | 39.24(3.8) | 83.50(1.4) | 50.62(2.8) | 50.43(5.2) | 79.24(1.0) | 59.08(2.1) | 74.21(1.7) | 70.90(12.6) | 4.00(4.0) | 60.60(2.2) | 23.20(2.7) | 86.31(29.5) |
| Naive Bayes | 45.88(2.5) | 78.79(1.4) | 49.36(2.7) | 39.55(1.8) | 79.48(0.7) | 63.57(2.7) | 78.49(2.6) | 60.46(1.2) | 45.33(7.2) | 78.46(3.1) | 49.95(7.5) | 42.21(6.7) |

TabPFN v2 on ISP: 20-model ensemble with 10k subsamples each. ConTextTab on ISP: 1,024-row stratified context per fold.

## B. Interpretability, Ablation and Deployment Pipeline

### SHAP Ablation Analysis: Purpose and Method

This section reports ablation experiments that serve two purposes. First, they validate the SHAP interpretability framework: if removing the features SHAP identifies as most important causes a large performance drop, SHAP rankings are genuinely reflecting predictive performance rather than spurious correlation. Second, they translate model explanations into directly actionable business intelligence, identifying the dominant churn driver in each industry sector.

For each dataset, the two features assigned the highest mean absolute SHAP values under TabICL v2 are removed, and the model is re-evaluated under the same 5-fold protocol. The drop in ROC-AUC and F1 measures the predictive contribution of those features. This methodology builds on the principle that an explanation tool that identifies genuinely important features should produce measurable performance degradation when those features are withheld.

### Results and Business Implications

Table D presents the ablation findings. The drops are substantial and consistent. Removing just two features causes ROC-AUC reductions of up to 17.0 pp (Bank Customer) and F1 reductions of up to 19.7 pp, confirming that SHAP correctly identifies load-bearing features rather than peripheral ones. These results have direct managerial value:

*Table D.* Ablation results: TabICL v2 performance drop when both top SHAP-identified features are removed. Large drops confirm that SHAP rankings reflect genuine predictive importance.

| **Dataset** | **Top Feature** | **2nd Feature** | Δ ROC-AUC | Δ F1 |
|---|---|---|---|---|
| Bank Customer | Age | NumOfProducts | −17.0 pp | −19.7 pp |
| Fitness Club | Avg_class_freq | Lifetime | −10.5 pp | −12.9 pp |
| IBM Telco | Contract | Dependents | −1.9 pp | −2.0 pp |
| Telecommunications | Cust. svc calls | Intl. plan | −13.6 pp | −16.1 pp |
| Credit Card | Total_Trans_Ct | Total_Trans_Amt | −7.0 pp | −7.1 pp |
| IBM HR Employee | OverTime | StockOptionLevel | −6.6 pp | −8.2 pp |
| ISP | remaining_contract | bill_avg | −7.5 pp | — |
| E-Commerce | Tenure | Complain | −5.1 pp | −5.7 pp |
| Healthcare Patient | Overall_Sat. | Days_Since_Visit | −5.2 pp | −3.5 pp |

**End-to-End Deployment Pipeline**

TabICL v2 is the strongest overall prediction model in our benchmark, and SHAP is the validated explanation layer. The pipeline is designed to be readily reusable across datasets. An analyst imports any customer dataset into the provided code, follows the five stages shown in Figure A, and receives a scored, explained, and actionable at-risk list within a single session.

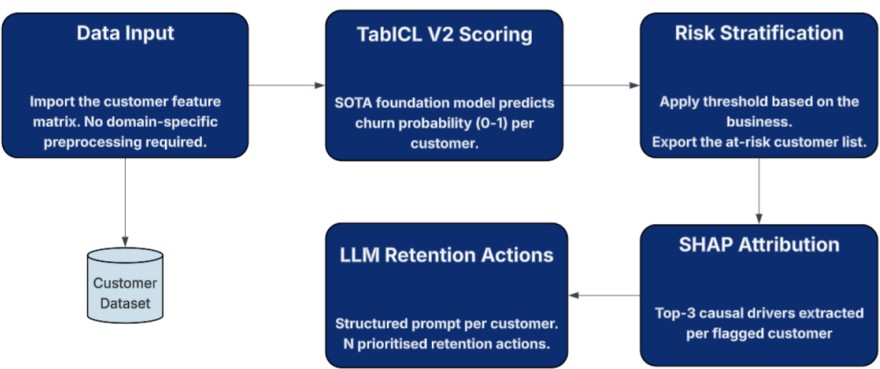

*Figure A.* End-to-end churn prediction and action pipeline.

## C. Model Architecture and Characteristics

Table E provides a comparative overview of all TFM variants evaluated. The models span four distinct pretraining paradigms (synthetic SCM, synthetic mixed, real-world, and real+semantic), two attention mechanisms, and a 50-fold range in parameter count. Full architecture diagrams follow the table with extended descriptions.

*Table E.* Characteristics of all tabular foundation model variants evaluated. *Synthetic*: pretrained on algorithmically generated datasets. *Real*: pretrained on curated real-world table corpora. Context limit: native maximum in-context training rows without approximation.

| Model | Params | Pretraining Data | Architecture | Novel Contribution | Context Limit | License |
|---|---|---|---|---|---|---|
| TabPFN v2 | ~93M | Synthetic (SCM/DAG) | Cell-level 2D attention | Alternating row/col attention; SCM priors | ~10K rows | Apache 2.0 |
| TabICL v1 | ~23M | Synthetic (SCM + tree) | Two-stage col→row ICL | Column-token architecture; tree priors | ~500K rows | Apache 2.0 |
| TabICL v2 | ~95M | Synthetic (novel engine) | Two-stage + scalable softmax | QASSMax; repeated feature grouping | ~500K rows | Apache 2.0 |
| TabDPT | ~100M | Real (123 datasets) | FAISS retrieval-augmented | Self-supervised masking; real-data scaling | Unlimited | Research |
| Mitra | 72M | Synthetic (mixed priors) | 12-layer Transformer | Principled prior mixture (SCM + tree-based) | ~500K rows | Apache 2.0 |
| LimiX-2M | 2M | Synthetic (hierarchical) | Asymmetric dual attention | Scale study; hierarchical difficulty prior | ~500K rows | Apache 2.0 |
| LimiX-16M | 16M | Synthetic (hierarchical) | Asymmetric dual attention | Scale study; retrieval ensemble at 16M | ~500K rows | Apache 2.0 |
| ConTextTab | ~70M | Real (T4/TabLib corpus) | Table-native ICL + col-name | Semantic column-name encoding (MiniLM) | 8,192 rows | Apache 2.0 |

## TabPFN v2 (Hollmann et al., 2025)

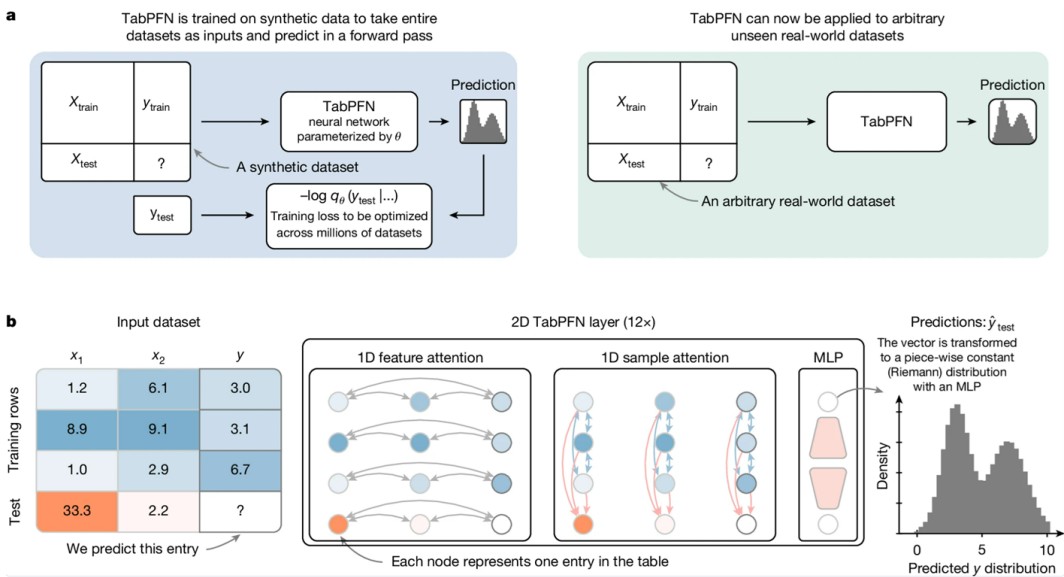

*Figure B.* TabPFN architecture: offline synthetic prior-fitting (left) and inference attention mechanism (right). Source: Hollmann et al. (2025).

TabPFN is a Prior-data Fitted Network trained to approximate the posterior predictive distribution for tabular classification. It represents each table as a set of (row, feature) cell tokens and applies alternating row-wise and column-wise Transformer attention, enabling the model to capture cross-feature interactions without any dataset-specific parameter updates. Pre-trained on millions of synthetic datasets generated from structural causal models (SCMs) and Bayesian neural networks, the v2 release extended the original to handle up to 10,000 training samples, 500 features, and both classification and regression. Its cell-based tokenisation scheme is architecturally unique among TFMs: encoding each cell individually preserves fine-grained feature interactions that row-level encoders aggregate away. A practical constraint is its $O(n^2m + nm^2)$ attention complexity, making it computationally intractable for the ISP dataset without a 20-model subsampled ensemble.

## TabICL v1 & v2 (Qu et al., 2025; 2026)

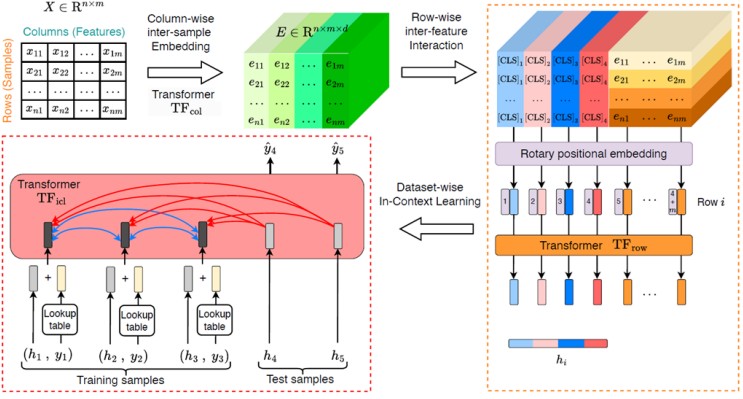

*Figure C.* TabICL v2 architecture: repeated feature grouping, target-aware embedding, and QASSMax scalable attention. Source: Qu et al. (2026).

TabICL decouples column-wise embedding from row-wise in-context learning, enabling linear scaling with row count while preserving cross-feature attention. Version v2 adds three major innovations: repeated feature grouping (circular shifts

of feature columns that preserve granularity while reducing sequence length), target-aware embedding (injecting label information directly into training tokens), and QASSMax (a query-aware scalable softmax that generalises attention to datasets of arbitrary size). Trained with a novel diversity-maximising synthetic data engine and the Muon gradient optimiser instead of AdamW, TabICL v2 ranked first on the TabArena public leaderboard at publication. Its most operationally significant advantage over all other TFMs in this benchmark is zero inference-time constraints: it processed all nine datasets in full, including the 72,274-row ISP dataset, in a single forward pass without subsampling, retrieval caps, or fine-tuning limits.

## TabDPT (Ma et al., 2025)

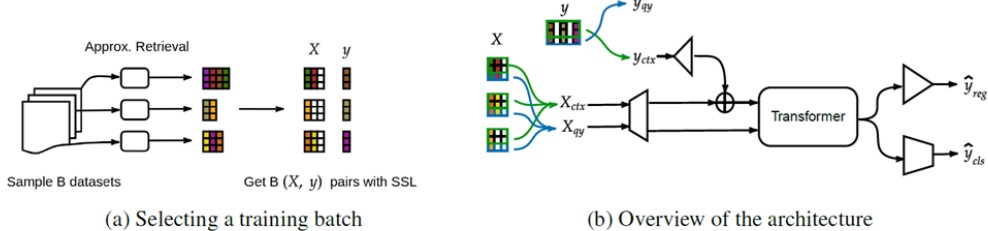

(a) Selecting a training batch          (b) Overview of the architecture

*Figure D.* TabDPT architecture: context and query rows are embedded and fused via a Transformer encoder; retrieval selects the $K$ nearest rows at inference via FAISS. Source: Ma et al. (2025).

Unlike all other TFMs in this benchmark, TabDPT is pre-trained exclusively on real-world datasets rather than synthetically generated data. It applies self-supervised masked-column prediction across 123 curated real-world tables (approximately 2 billion cells), learning transferable row-level representations without any labelled supervision during pre-training. At inference, FAISS nearest-neighbour retrieval identifies the $K$ training rows most similar to each test query and passes them as context, effectively constructing a task-specific in-context window from the real training set. This retrieval mechanism directly tests whether real-data pre-training generalises to industry churn data better than synthetic priors, and represents a fundamentally different inductive bias from the SCM-based models. In this benchmark, TabDPT's retrieval budget was constrained to $K=256$ (vs. the default $K=2,048$) due to GPU memory limits, likely understating its full potential performance.

## Mitra (Zhang et al., 2025)

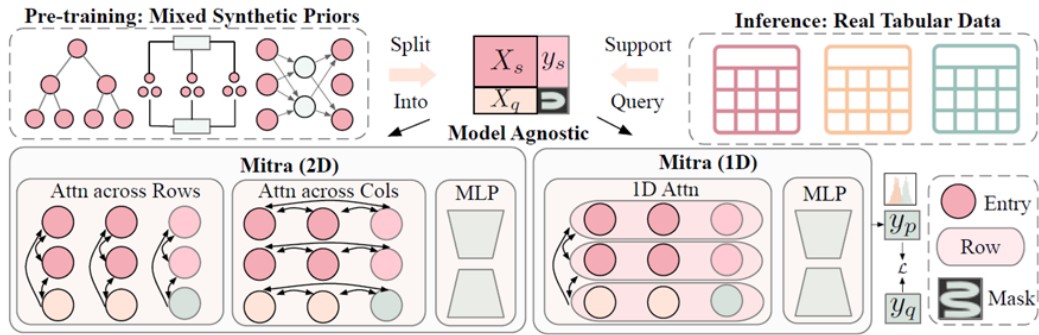

*Figure E.* Mitra: the Generalizability Matrix $\mathbf{G}$ and Performance Vector $\mathbf{P}$ guide selection of a diverse, distinctive prior mixture. Source: Zhang et al. (2025).

Mitra's primary contribution is a principled framework for analysing and selecting synthetic data priors for TFM pre-training. It characterises each candidate prior type using a Generalizability Matrix $\mathbf{G}$ (measuring cross-prior generalisation via AUC) and a Performance Vector $\mathbf{P}$ (measuring real-world accuracy), then selects a mixture that jointly maximises performance, diversity (high off-diagonal $\mathbf{G}$ variance), and distinctiveness (low off-diagonal entries). The resulting mixture combines structural causal model priors and tree-based priors (gradient boosting, random forest, decision tree, extra trees), whose

complementary decision boundary geometries provide broader coverage of real-world tabular distributions than either prior type alone. Built on the same 2D cell-attention architecture as TabPFN v2, Mitra demonstrates that prior design is equally important as architectural innovation in determining TFM performance. In this benchmark, Mitra achieves the highest F1 and PR-AUC on the Telecommunications dataset, confirming that its mixed prior captures threshold-structured churn patterns that purely SCM-based models miss.

**LimiX 2M & 16M (LimiX Team, 2025)**

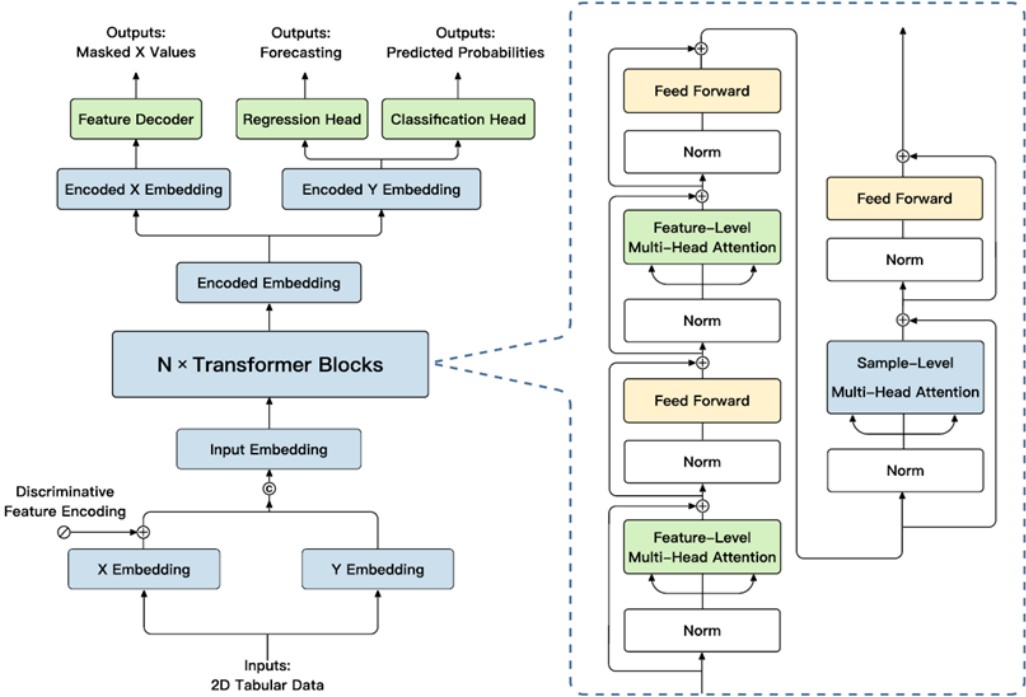

*Figure F*. LimiX-16M: asymmetric dual feature-level and sample-level attention blocks produce classification, regression, and masked-value output heads. Source: LimiX Team (2025).

LimiX treats each table as a joint distribution over observed features and missing values, embedding both feature values (X embedding) and target labels (Y embedding) through a shared discriminative feature encoder before passing them into Transformer blocks. Each block applies two feature-axis attention passes followed by one sample-axis attention pass, an asymmetric configuration empirically shown to capture inter-feature relationships more efficiently than symmetric designs. The 2M and 16M variants share this architecture but differ in capacity: the 16M model benefits from retrieval-augmented ensembling at large context scales, while the 2M model exhibits better calibration on smaller datasets. LimiX's hierarchical synthetic prior introduces controllable difficulty levels during pre-training, from linearly separable to highly non-linear, producing a prior distribution that covers a wider range of real-world decision boundaries than fixed-difficulty sampling. In this benchmark, LimiX-16M achieves the best overall F1 and ROC-AUC on Bank Customer Churn, and the ISP dataset runs used a 40,000-row context cap per fold due to hardware memory limits.

**ConTextTab (Spinaci et al., 2025)**

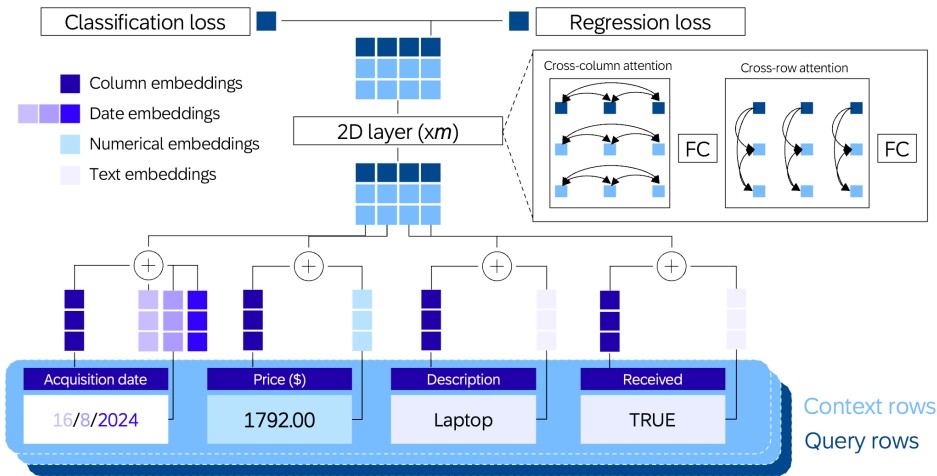

*Figure G.* ConTextTab: per-cell embeddings (column headers, dates, numerics, text) are summed and processed by interleaved cross-column / cross-row attention. Source: Spinaci et al. (2025).

ConTextTab bridges table-native ICL and natural language understanding by encoding column header names as semantic signals alongside numerical cell values. For each column, the cell value is processed by a standard numerical embedding layer while the column name is separately encoded by a frozen sentence encoder (all-MiniLM-L6-v2) and summed into the cell representation, allowing the model to exploit naming conventions as a weak prior without requiring free-text cell content. Pre-trained on the T4 dataset (a curated subset of the TabLib corpus of over one million real-world tables), it is the only TFM in this benchmark that combines real-data pre-training with semantic column-level context. This semantic enrichment provides a meaningful distributional prior even for purely numerical datasets: a column named monthly_charges carries structural associations that a model pre-trained only on synthetic data must infer entirely from observed numerical patterns. In this benchmark, ConTextTab was evaluated with max_context_size=1024 and bagging=2 on a Tesla T4 GPU; it achieves competitive ROC-AUC across all nine datasets but exhibits more variable F1 on severely imbalanced datasets due to limited positive-class representation within the 1,024-row context window.

## D. Dataset Links

- Bank Customer Churn: https://www.kaggle.com/datasets/shrutimechlearn/churn-modelling

- Fitness Club Churn: https://www.kaggle.com/datasets/adrianvinueza/gym-customers-features-and-churn

- E-Commerce Churn: https://www.kaggle.com/datasets/ankitverma2010/ecommerce-customer-churn-analysis-and-prediction

- IBM Telco Churn: https://www.kaggle.com/datasets/blastchar/telco-customer-churn

- Telecommunications Churn: https://www.kaggle.com/datasets/mnassrib/telecom-churn-datasets

- Healthcare Patient Dropout: https://www.kaggle.com/datasets/nudratabbas/patient-churn-prediction-dataset-for-healthcare

- ISP Churn: https://www.kaggle.com/datasets/mehmetsabrikunt/internet-service-churn

- Credit Card Customer Attrition: https://www.kaggle.com/datasets/sakshigoyal7/credit-card-customers

- IBM HR Employee Attrition: https://www.kaggle.com/datasets/pavansubhasht/ibm-hr-analytics-attrition-dataset

