# OpenReview forum: "Benchmarking Tabular Foundation Models for Churn Prediction"
_ICML.cc/2026/Workshop/FMSD — FMSD @ ICML 2026 Poster_

### Official Review · Reviewer_iwRx · 2026-05-18
**A Cross-Industry Benchmark of Tabular Foundation Models for Churn Prediction**

**Rating:** 6
**Confidence:** 4

**Review:**

## Summary
Cross-industry benchmark of 16 models across nine churn datasets, arguing that TFMs outperform classical, deep learning, and tree ensemble baselines under zero-shot conditions.

## Strength
- breadth of evaluation: the benchmark covers 9 datasets across various sectors, with variation in size (1.5K-72K rows), dimensionality (9-30), and class imbalance (~15% to ~68%).
- justified choice of metrics: PR-AUC as primary metric is well-justified for imbalanced churn data.

## Areas for Improvement
- hardware constraints for TFMs: Experiments run on a T4 (16 GB VRAM), which in my opinion is rather small given modern compute. Conclusions about which TabICLv2 is "state-of-the-art" are hardware-conditional. Using slightly more compute (e.g. with ~50GB VRAM) might show a different landscape.
- TFMs perform rather similarly, with no significant testing: In Table 2, it seems the top TFMs performance are still very much clustered. I find directly claiming TabICLv2 as the leading TFMs among these rather unsupported and statistically insignificant.
- the SHAP ablation seems redundant, and with inaccurate claim: the authors identify top SHAP features, remove them, and show performance drops -- this is a standard SHAP sanity check, not an independent finding. More importantly, the paper frames this as establishing "causal drivers," which it does not. Predictive relevance != causality. The section should be revised accordingly.
- XGBoost config is underspecified. It's unclear whether XGBoost is tuned or run at defaults. An untuned XGBoost inflates the apparent TFM advantage.
- lack actionable insights as the authors claim to have: XGBoost + SHAP pipeline is not something new in the industry.

## Detailed Comments
- n/a

## Justification of Score
- the question is practically relevant
- the empirical setup is still solid, although with much rooms for improvement
- overstated interpretability ablation (it's more of a sanity check)
- statistical rigor can be improved

---

### Official Review · Reviewer_tv93 · 2026-05-20
**Strong cross-industry TFM benchmark for churn prediction**

**Rating:** 8
**Confidence:** 4

**Review:**

### Summary

This paper benchmarks tabular foundation models for customer churn prediction across nine public datasets spanning seven industry sectors. It compares sixteen models, including classical baselines, neural tabular models, tree ensembles, and several recent tabular foundation models. The main result is that TFMs, particularly TabICL v2, achieve the strongest overall performance, with PR-AUC used as the primary metric due to class imbalance. The paper also includes interpretability analysis using SHAP, LIME, EBM, and feature-ablation studies, and translates some predictive gains into estimated financial impact.

### Strengths

This is a strong and relevant benchmark paper. Customer churn prediction is a practically important structured-data use case, and the paper fits the workshop scope very well.

The experimental setup is broad and well motivated. The authors evaluate nine datasets across banking, fitness, e-commerce, telecommunications, healthcare, internet services, and HR, with substantial variation in sample size, feature count, and churn rate. This makes the empirical claims much more credible than a single-dataset comparison.

The model selection is strong. The benchmark includes sixteen models across classical methods, neural tabular architectures, tree ensembles, and multiple recent TFMs, including TabPFN, TabICL, TabDPT, Mitra, LimiX, and ConTextTab. This is well aligned with the workshop’s focus on modern foundation models for structured data.

The choice of PR-AUC as the primary metric is appropriate and justified. Since churn datasets are often imbalanced and missing true churners is costly, PR-AUC is a sensible primary metric alongside F1, ROC-AUC, and log loss.

The results are clearly presented, and the comparison is informative because strong non-foundation baselines such as XGBoost and Random Forest remain competitive on several datasets while TFMs show consistent gains overall.

The interpretability and ablation components strengthen the paper. The authors compare SHAP, LIME, and EBM explanations, and then test whether removing top SHAP features reduces performance.

The appendix is strong, with full performance tables, model-characteristic summaries, architecture descriptions, dataset links, and ablation results.

### Areas for Improvement

The main weakness is that some claims are too strong for the evidence provided. Phrases such as “fully industry-ready”, “businesses across all sectors can embrace today”, and “customer churn is a solvable problem” overstate what can be concluded from public Kaggle-style benchmarks. The results support the claim that TFMs are promising and often outperform strong baselines on these datasets, but deployment readiness would require additional evidence around temporal drift, calibration, intervention effects, retention-budget constraints, and real campaign outcomes.

The novelty claim should be qualified. The paper’s claim to be the first broad benchmark of modern TFMs for churn prediction appears plausible. However, the broader statement that “no prior benchmark spans multiple industry sectors simultaneously” seems too strong. For example, Haddadi et al. (2024) evaluate customer churn prediction on three public imbalanced datasets spanning telecommunications, online retail, and banking, using fourteen classification methods and resampling strategies. The paper’s novelty would be more accurate if framed as the first broad cross-industry benchmark of recent tabular foundation models for churn prediction.

The interpretability section should avoid causal language. The SHAP and ablation analyses show that the selected features are predictive and load-bearing for model performance, but they do not establish causality. Statements such as “causal drivers” or “causal structure” should be replaced with “predictive drivers”, “important features”, or “decision-relevant features” unless a causal identification strategy is added.

The paper should briefly discuss contamination and memorisation risk. Several datasets are public Kaggle datasets, while some TFMs are pretrained on real-world table corpora or guided by public benchmark performance. Since structured-data corpora are small and often reused, a short contamination discussion would be valuable and aligned with the workshop’s emphasis on contamination-aware benchmarking.

### Detailed Comments
- Please qualify the novelty claim. The contribution is strong as a broad cross-industry benchmark of modern TFMs for churn prediction, but prior multi-sector churn benchmarks do exist, including Haddadi et al. (2024), which evaluates churn prediction across telecommunications, online retail, and banking datasets.
- Please soften deployment claims. The results support TFMs as promising tools for churn prediction, but “fully industry-ready” and “businesses across all sectors can embrace today” are stronger than the evidence warrants.
- Please replace causal language in the interpretability and ablation sections. The ablation results support predictive importance, not causality.
- Please add a short contamination-risk discussion, especially for public Kaggle datasets and TFMs pretrained on real-world table collections.
- Consider reporting more complete explanation outputs in the appendix, such as per-dataset SHAP summaries beyond the top two features and LIME results. This is minor, but it would further strengthen the interpretability contribution.

### Justification of Score

This is a clear, relevant, and well-executed benchmark paper. It addresses an important real-world structured-data problem, evaluates a broad and appropriate model set, uses a well-motivated metric, presents results clearly, and includes useful interpretability and ablation analysis. The main issues are mostly about overclaiming and claim precision rather than core technical weaknesses. I would support acceptance, especially if the authors qualify the novelty/deployment claims and add a short contamination discussion.

Reference: Haddadi et al. (2024), *Customer churn prediction in imbalanced datasets with resampling methods: A comparative study*, Expert Systems with Applications.

---

### Official Review · Reviewer_NiSe · 2026-05-21
**Review of Benchmarking Tabular Foundation Models for Churn Prediction**

**Rating:** 7
**Confidence:** 3

**Review:**

## Summary
Authors present a comprehensive performance comparison of various churn prediction models on nine datasets from various domains. There are 6 distinct tabular foundation models (some with several versions) and multiple baselines covering neural, tree and classical methods. The authors show that foundation models are consistently outperforming the baselines, and that in particular TabICLv2 stands out as being a good candidate for a state-of-the-art-defining model. Authors also supply an interpretability and ablation study.

## Strengths
* A huge set of models, baselines, datasets and domains. Also multiple evaluation metrics. Overall this makes an impressive analysis.
* Section 5 with interpretability and ablation study, showing relevant conclusions,
* Focus on business value brought by improving churn predictions (more on that later).
* Nice little overview of TFM architectures in Appendix C.

## Areas for Improvement and Suggestions
* It is a bit too much of an effort for the reader to validate the claims made in the text with result tables, as many of them are moved to the appendix and they're overall pretty huge. My proposal would be to rather show one metric for all datasets in the main part (as different metrics are anyway largely correlated and just bloat the tables; the text focuses mostly on PR-AUC anyway so why not show that), and move the other metrics to the appendix.
* I am really not a fan of reporting max-type results, especially in the abstract. The authors report "up to 9.23pp" over XGBoost, but it's clear from figure 1 that such a large improvement is there only for 2 out of 9 datasets. Being perhaps a bit too sarcastic, one could ask if you would also write this if 8 datasets were worse than XGBoost, but one was 9pp better? Sarcasm aside, I'd rather propose something like a mean improvement, especially in the abstract, which is the most exposed part. Looking at Fig. 1, I think the number will still be respectable.
* Also, when XGBoost is your main point of reference, an obvious question is how well it was tuned on each of the 9 datasets. It would be nice to discuss this at least in the Appendix. I am aware that the lack of having to do a major fine tuning procedure for TFMs is a major advantage of the approach, but for a fair accuracy comparison I guess you want to compare the performance also against a proper hyperparam-tuned tree model (to a lesser extent this also applies to the other baselines).
* I am not sure if section 2.2 (Interpretability Methods) brings any value at all, especially given that it's largely repeated in Section 5. I'd rather drop it and move parts of it inside Section 5 if any are missing there.
* I am really skeptical about the numbers in the business value calculation in Section 1 (Introduction). You do another one in section 4.3 and that one's much more convincing, e.g. including the 30% retention success -- while I think that the calculation in the intro assumes 100%? Also, it would be nice to have some reference for the 30% as a conservative estimate. I'd be tempted to drop the back-of-envelope calculation in Section 1 and rather put some aggregate monetary quote from Sec. 4.3 and refer the reader to it for details.
* Last tiny comment: "Customer churn is a solvable problem" <-- this sentence at the beginning of Section 6 (Conclusions) would make a much better impression if it contained a word "prediction" or "forecasting" somewhere in it.


# Justification of Score
A solid comprehensive analysis with practical industry consequences. Most of my critical comments are about the form of the manuscript rather than its factual content. One thing that stopped me from rating the paper even higher is the point about missing information on how well XGBoost is tuned, as consistently beating an expertly-tuned XGBoost on all datasets is much more impressive than beating an out-of-the-box XGBoost, and I'm not really sure what are we looking at in the text.